# Super-Resolving Methodology for Noisy Unpaired Datasets

**DOI:** 10.3390/s22208003

**Published:** 2022-10-20

**Authors:** Sung-Jun Min, Young-Su Jo, Suk-Ju Kang

**Affiliations:** Department of Electronic Engineering, Sogang University, Seoul 04107, Korea

**Keywords:** super resolution, unpaired dataset, average denoising

## Abstract

Although it is possible to acquire high-resolution and low-resolution paired datasets, their use in directly supervised learning is impractical in real-world applications. In the present work, we focus on a practical methodology for image acquisition in real-world conditions. The main method of noise reduction involves averaging multiple noisy input images into a single image with reduced noise; we also consider unpaired datasets that contain misalignments between the high-resolution and low-resolution images. The results show that when more images are used for average denoising, better performance is achieved in the super-resolution task. Quantitatively, for a fixed noise level with a variance of 60, the proposed method of using 16 images for average denoising shows better performance than using 4 images for average denoising; it shows 0.68 and 0.0279 higher performance for the peak signal-to-noise ratio and structural similarity index map metrics, as well as 0.0071 and 1.5553 better performance for the learned perceptual image patch similarity and natural image quality evaluator metrics, respectively.

## 1. Introduction

In visual inspection tasks involving semiconductor images, resolution is an important parameter that determines the structural information or defects. In such tasks, images are often acquired at lower resolutions owing to the limited resources of sophisticated hardware, and inspection is not possible because of lack of detail and texture. Therefore, the super-resolution (SR) technique is frequently used for restoring low-resolution (LR) to high-resolution (HR) images to enhance details and structural information in practical visual inspection tasks.

Generally, image restoration (IR) aims to restore high-quality images from degraded images, but it is an ill-posed problem for an observed image and can be modeled as follows:(1)y=DHx+n,
where *x* is the original input image, *D* is the down-sampling operator, *H* is a blur kernel, and *n* is additive noise. SR tasks constitute a field of IR that has been actively researched and aims to restore HR images containing missing high-frequency details from LR images. Recently, deep-learning-based SR methods have shown improved performance compared to existing methods from a practical point of view.

However, most existing SR methods assume that both the input and reconstructed images are noise-free and aligned perfectly. Unfortunately, in practical cases, noise is inevitably included during image acquisition, making this assumption invalid for real-world applications and rendering these SR tasks more difficult. This is because it is difficult to prevent noise amplification during up-sampling, which often leads to loss of information and emergence of artifacts. Further, alignment problems occur when acquiring HR–LR image pairs. As observed in [1,2], when an HR–LR pair is directly photographed and the alignments do not match, scale-invariant feature transform (SIFT) and random sample consensus (RANSAC) methods are applied for rough alignment matching, and various losses are applied for fine alignment matching to train the model.

Therefore, SR models for visual inspection should be considered in cases such as scanning electron microscopy (SEM) semiconductor datasets that have a different domain than typical images. However, since there are no public datasets available as our target, we present an SR model training solution for cases where actual HR–LR pairs are obtained. Therefore, HR–LR pairs were generated as a dataset under several assumptions for practical image acquisition conditions, and SR training was conducted with this dataset.

First, the HR and LR images are both noisy, and in the LR case, the noise level is large, so detailed information is insufficient. Second, the HR–LR images are misaligned under the assumption that the pairs are photographed; thus, training proceeds in an unsupervised manner. Therefore, we configured the HR–LR unpaired dataset with a public SEM dataset [3] and generated heavy-noise LR images to match the actual acquisition environment. Each noisy LR image was acquired several times and averaged to obtain a single LR image with averaged noise. Furthermore, for the unpaired HR–noisy-LR images, we generated aligned HR and LR image pairs with averaged noise through the noise extraction and injection method. This paired dataset was used to assess the effectiveness of several existing SR models. In summary, we present a data preprocessing methodology for unpaired heavy-noise SR task herein. The overall architecture of the proposed method is depicted in Figure 1 and the main contributions of this work are as follows.

We propose a general scheme in unsupervised super resolution for heavily noised dataset that is simple yet effective and can be applied in overall SR networks.**Average Denoising**: We obtain multiple heavy-noise LR images that are equal to taking multiple images in real camera shooting conditions for visual inspection tasks and average them, which reduces the noise level while preserving structural and visual details.**Paired Data Generation**: Using the noise extraction and noise injection approach introduced in RealSR [4], we generate a paired dataset while maintaining the noise characteristics of the source dataset.Through thorough experiments, we show the effectiveness of the proposed scheme, which shows better performance in several evaluation metrics widely used in image quality assessment.

## 2. Related Work

Even prior to learning-based methods using spatial and frequency domains, SR was studied in various fields such as medical image processing and aerial imaging to achieve images with better quality [5]. However, with the advent of learning-based methods, supervised learning using paired datasets has achieved great performance. Thus, various methods have been proposed to improve the restoration performance from LR to HR using SR networks based on convolutional neural networks (CNNs). Since the introduction of the first learning-based models [6], early studies [7,8] have found that deeper networks perform better with residual learning. The residual channel attention network (RCAN) [9] achieved further improvements in depth and performance. Up-scaling strategies have also been studied in the past, with iterative up-scaling and down-scaling of deep back-projection networks (DBPNs) for super resolution [10]. Using both perceptual and adversarial losses for the generative adversarial network (GAN) structure, the perceptual quality was improved in SRGAN [11], and its enhanced version ESRGAN [12] has been used widely. Following this, super resolution has been adopted for various applications such as medical image [13] and face hallucination [14].

### 2.1. Unsupervised/Unpaired Super Resolution

The aforementioned SR models produce poor results when the inputs are real-world LR images because they are trained on paired data for which the LR images are generated by simply down-sampling the corresponding HR images with a bi-cubic kernel. To solve this problem, some studies [1,15] directly collected HR–LR pairs using specific camera settings and introduced real-world paired datasets. Collecting paired data is a laborious task that requires composing a dataset by simultaneously capturing non-moving objects as pairs. To overcome this problem, several practical SR approaches have been proposed, in which the GAN structure is used to learn the conditional distribution of the LR image domains given HR images [16].

As one of the methods of handling unsupervised SR tasks, either the source or target domain is chosen, as in [17], and it is considered a challenge to proceed with SR. Since the target-domain HR image must be restored from the corresponding source-domain LR image, it is essential to generate an aligned paired dataset. Therefore, RealSR [4], the winner of the NTIRE 2020 challenge, proposed the design of a new degradation framework for real-world images by estimating various blur kernels as well as the actual noise distribution.

### 2.2. Multitask Methods for Denoising and Super Resolution

As an initial approach to handling noisy data, sparse representation schemes [18,19] were used to reconstruct images. By formulating the IR task as an inverse problem, the IR data terms are unique for each objective. Therefore, more generic functions that can be applied to multiple tasks are used in selection of the prior, where a denoiser is implemented as a regularizer [20,21], and the so-called plug-and-play prior is employed [22]. Furthermore, several approaches have been considered to combine denoising with SR models [23]. The authors of [24] proposed denoising and SR tasks to reconstruct a noiseless image using an interpolation approach with a local fractal dimension.

Aside from the conventional SR tasks, BSRGAN [25] was proposed as a real-world degradation model to negate the downsides of synthetic data generation and build a model robust to different combinations of down-sampling kernel, blur kernel, and noise. This approach showed excellent performance with real-world datasets, for which an unknown degradation model is expected. However, since the models trained with image pairs generated by such real-world degradation models are considered for the general case, it is confirmed that denoising was achieved above the required level. Therefore, the real-world degradation model is not suitable for noise-specific visual inspection tasks with fixed HR and LR noise levels.

Hence, joint-type denoising and SR-type networks or real-world degradation models have poor ability to preserve texture details and structural information for severely degraded noisy images that lack flexibility (trade-off) with respect to denoising degree and detail preservation. Therefore, we consider an average denoising method rather than an algorithmic approach to heavy noise, thereby enabling both noise reduction and detail preservation.

## 3. Data Formulation and Methods

Figure 2 shows the limitation of naive application of existing SR networks. Most existing SR networks deal with little additive noise. However, in special cases, such as heavily noised conditions, these methods suffer from loss of information due to degradation and fail to output satisfactory results, as they have blurry edges and lack of structural details. To deal with this problem, we synthetically formulated a dataset and applied average denoising by generating a paired dataset using a noise injection method. A detailed explanation follows.

We assumed that multiple images were acquired with a camera in actual semiconductor settings that encompass general real-world problems. Hence, a different approach is required than the conventional SR task that utilizes traditional HR–LR pairs. There are three main problems with the semiconductor image dataset. First, unlike the generally organized paired dataset, actual HR–LR pairs are not applicable for supervised learning directly owing to misalignments. Second, the existing SR algorithms address very low noise or even noise-free environments that have discrepancies in terms of acquiring real-world semiconductor images. Third, a public dataset that is in agreement with the problem formulation is not provided. Therefore, we propose a general scheme to create an unpaired heavy-noise dataset by generating data to overcome the above problems.

### 3.1. Unpaired Data Generation and Average Denoising

**Data Formulation:** We use the ground-truth dataset as the SEM dataset [3] to establish an environment similar to that of semiconductor image acquisition for visual inspection. Figure 1a shows the unpaired dataset generation process for problem formulation. After randomly shifting from 30 to 100 pixels from the ground-truth HR image, a clean LR image is generated by down-sampling. Usually, a noise variance of 60 to 90 is added to ensure a setting similar to real-world data acquisition. This sequence of intended misalignment with noise injection is performed to embody the characteristics of an unpaired dataset with heavy noise similar to real-world data acquisition conditions.

Hence, the additional noise comprised variances of 30, 60, and 90 to verify the effectiveness of average denoising for heavy noise. In proceeding with this heavy-noise SR task, noise reduction must essentially be applied to existing SR algorithms. However, the trade-off between noise and detail should be adjusted meticulously for visual inspection. When implementing conventional blind denoising algorithms, we tend to remove both the details and noise simultaneously, as was verified in the ablation study. Therefore, we solve this problem by applying an average denoising method that can be implemented easily at production facilities.

Unlike learning-based denoising algorithms, average denoising is a very practical and straightforward approach. By capturing several frames of noisy images and taking their average, our method reduces the effects of noise; this method assumes that the noise in the acquired image is uniformly distributed in a Gaussian manner. Therefore, we can preserve repeated edge or detail information while removing probabilistic noise.

Although there are disadvantages to using multiple input images, there are also clear advantages. The first drawback is that, compared to learning-based methods, the details are preserved and only noise is eliminated (especially when there is heavy-noise input). Second, the degree of denoising can be freely adjusted according to noise level by determining the number of images that are averaged. We averaged 1, 2, 4, 8, and 16 images to confirm the noise reduction effects, and based on the results, we trained the SR model to confirm performance. Figure 3 shows averaged images with different numbers of input images as well as a clean LR image that is directly down-sampled from the HR image. The more images that we average, the better are the noise reduction effects and preservation of image details. However, we need to consider the trade-off between the number of images and the SR output image.

### 3.2. Paired Data Generation

Although our task considers unpaired dataset generation and heavy-noise reduction similar to that of a real image acquisition environment, the unsupervised method of the paired dataset has to be examined. To resolve this problem, the NTIRE challenge 2020 winner, RealSR [4], was used. Regarding the noisy images, we directly injected noise patches into the down-sampled image to create more-realistic LR images. Since the HR component is lost during down-sampled data generation, the degraded noise distribution also changes in the process. Therefore, we collected noise patches from the source dataset based on the following condition:(2)$$\sigma (n_{i})<var,$$
where σ(·) denotes the function to calculate variance, $n_{1},n_{2},\ldots ,n_{i}$ are a series of noise patches, and *var* is the max value of variance. When applying the paired generation method corresponding to Figure 4, the original HR image is first down-sampled, and the noise extracted from the average denoised image is injected. Through this process, the paired dataset is generated while maintaining the noise component of the LR image with the averaged noise from the unpaired dataset. This generated LR image is now paired with the HR image, and from now on, we mark this paired LR image as $LR_{average}$ for convenience. For example, by $LR_{average16}$, we denote that 16 is the number of averaged images. Accordingly, the paired dataset is applied to a general SR network for supervised learning.

Since the DPED [2] dataset introduced in the NTIRE challenge deals with very small noise, the noise patches can be determined by noise variance. However, there is difficulty in applying the above method to our task, which has a very large noise component. As the noise increases, it is ambiguous to determine the threshold of the noise patch variance because it is difficult to distinguish whether the reason for the large variance is due to noise or due to structural information such as edges. Hence, the threshold should be set clearly, because the noise variance patch is the determining factor in the algorithm. The noise patches for $LR_{average4}$ are shown in Figure 5. The blue boxes are the normal noise patches as expected, and the red boxes are the incorrectly extracted noise patches. As we increase the threshold variance value, patches with structural information are also extracted in addition to the noise patches. However, appropriate levels of noise patches have to be extracted since training would be challenging with considerable noise variance.

We can create these $LR_{average4}$, $LR_{average8}$, and $LR_{average16}$ images with relatively low noise levels. Even with very deliberate selection according to the variance thresholds, extracting only noise patches and generating $LR_{average1}$ and $LR_{average2}$ images is impossible since falsely detected patches are detected so frequently. Therefore, we were not able to conduct experiments on those average numbers of images because the corresponding paired HR–$LR_{average}$ dataset could not be acquired.

The degeneration kernel is also an essential factor for the degeneration model. Since we generated LR images using the bi-cubic down-sampling method, a further process with additional kernel estimation was unnecessary.

## 4. Results and Discussions

In this section, we first describe the datasets and implementation details. Then, we analyze the proposed scheme through qualitative and quantitative results. For fair comparison, we chose one of the state-of-the-art methods, SwinIR [26], as our baseline method. Finally, the effects on other SR networks, feasibility testing on average denoising, and evaluation of other denoising algorithms are described through ablation studies.

### 4.1. Dataset and Implementation

#### 4.1.1. Training and Test Datasets

As explained earlier, we used an SEM dataset [3] as the ground-truth images; the dataset comprises 4583 training and 215 test images. These images were used as our ground-truth HR images, and unpaired LR images were generated to construct the training and test datasets. In the case of the test dataset, we additionally generated shifted HR images that are paired with LR images, thereby enabling a ground-truth target image for the LR image. This is impossible under real data acquisition conditions but was achieved by our synthetic image generation as the purpose was for evaluation as a reference image. Therefore, full-reference evaluation metrics were applicable.

#### 4.1.2. Implementation Details

Training was conducted using the SwinIR network as the baseline for the paired dataset generated above, and its performance was confirmed. The training scheme followed that of SwinIR. As a classical SR model training for SwinIR, the Adam optimizer was used with a batch size of 128. The learning rate was initialized to $2e^{-4}$, and the window size was set to 64 for comparison with other networks. In the loss perspective, we implemented the naive $L_{1}$ pixel loss to be the same as in SwinIR.

### 4.2. Evaluation Metrics

We used peak signal-to-noise ratio (PSNR) and structural similarity index map (SSIM), which are commonly used for image restoration evaluations for generated data. These metrics focus more on image fidelity rather than on perfect visual quality. On the contrary, learned perceptual image patch similarity (LPIPS) [27] evaluates whether the visual features are similar; this metric uses the pretrained AlexNet [28] to extract image features and calculate the distance score between the two features. The natural image quality evaluator (NIQE) [29] is one of the non-reference metrics used for image quality assessments. Thus, higher PSNR and SSIM values indicate greater similarity with the ground-truth in terms of image pixels, whereas lower LPIPS distance and NIQE indicate better performance at a perceptual level.

### 4.3. Qualitative and Quantitative Results

Figure 6 shows the SR results using an average-denoised LR image. Figure 6b is the target HR image whose corresponding LR image is in Figure 6a, which is the aligned dataset generated for evaluation during the test phase. Figure 6c,e,g show $LR_{average4}$, $LR_{average8}$, and $LR_{average16}$, respectively. Figure 5d,f,h show the output SR images trained on the paired dataset generated by the proposed scheme. Comparing Figure 6a with Figure 6c,e,g, the effectiveness of average denoising is confirmed. While almost no details are found in Figure 6a, structural and texture details are seen in Figure 6g. Correspondingly, the SR results show more conceivable results with fewer crumpled areas and sharper edges.

Figure 7 shows the SR results using an average denoised LR image with several noise levels applied. For different noise levels, we trained the network using the generated HR–$LR_{average}$ paired dataset and tested on the shifted HR–$LR_{average}$ images. As the number of images used for average denoising increases, the noise in the restored image decreases, and the edge and texture details are restored well. This effect can be observed at higher noise levels, such as noise variances of 60 or 90.

Unlike the collapse of structural coherency or texture details in restored SR images trained using $LR_{average4}$ images, the SR results using $LR_{average16}$ images could reconstruct the structural and texture details well. This tendency is clearly observed in Figure 7b. Figure A1b in Appendix A shows larger errors as the noisy LR image contains noise-like details that are easily corrupted by random noise; hence, it is easier to compare the image patches. While the details are hardly observed in the SR results using $LR_{average4}$ images, speckle details are easily spotted in the SR results using $LR_{average16}$ images. For easier comparison, the error map is multiplied by a factor of five. For additional evaluations on test images, Figure A2 and Figure A3 are included in Appendix A.

Table 1 shows the quantitative results for Figure 7. We evaluated the effectiveness based on PSNR, SSIM, LPIPS, and NIQE, as noted earlier in Section 4.2. It is seen that as the number of averaged denoised images increases for each noise level (variance with 30, 60, and 90), the denoising effects is enhanced, leading to better performance in the SR task.

Through both qualitative and quantitative evaluations, it is shown that the proposed scheme is effective at handling unpaired heavy-noise SR tasks. This method is particularly effective in heavy-noise situations, such as with noise variances of 60 and 90, where almost no details are preserved in single LR images.

## 5. Ablation Study

### 5.1. Evaluations with Several SR Networks

To evaluate the proposed scheme, we experimented with the same sequence of images on different SR models, such as RCAN [9], ESRGAN [12], and MSRResNet [11]. To confirm their tendencies, we fixed the noise level variance to 60 and used $LR_{average16}$ as the input. All networks were trained with a patch size of 64. Figure 8 shows the qualitative comparison, and Table 2 shows the quantitative comparison. From Table 2, MSRResNet [11] shows the best results out of the three comparison models in terms of averaging noisy inputs, followed by RCAN [11]. However, we conclude that the effects of averaging multiple noisy inputs are feasible since the qualitative values of PSNR and SSIM increase as the averaging number increases. Furthermore, our proposed concept may be applicable to other general SR models and even transform-based networks, such as SwinIR [26].

### 5.2. Evaluations with Several Denoising Algorithms

This section explains why we used average denoising instead of conventional denoising algorithms. As benchmark methods, FFDNet [32], SwinIR [26], and Restormer [33] were used, and the comparison results with average denoising are shown in Figure 9. It is confirmed that Figure 9c,d lack noise removal, and that Figure 9e shows great performance for noise removal. However, at the same time, it is also confirmed that there is a trade-off, such as deletion of texture information. To avoid these problems, average denoising was selected, because such problems can have a critical impact on performance when used further for other tasks such as anomaly detection and segmentation in the future.

### 5.3. Feasibility Test of Average Denoising

We constructed an unpaired dataset with 1, 2, 4, 8, and 16 averaged images and experimented only on 4, 8, and 16 averaged images, which can be restored with paired data. Since the noise patch extraction and noise patch injection techniques of RealSR [4] were not available, performance verification for the $LR_{average1}$ and $LR_{average2}$ images could not be generated. Therefore, to verify the performance of the proposed denoising scheme, SR was conducted using a perfectly aligned dataset through average denoising. This dataset was generated only by average denoising and without noise extraction or noise injection. In real-world applications, these perfectly aligned HR–LR pairs are nearly impossible to acquire. Thus, this experiment was conducted to verify the effectiveness of average denoising.

Figure 10 and Table 3 show the qualitative and quantitative results, respectively, for the aligned image pairs. As expected, the averaged images using 16 noisy LR images showed the best performance among other images. The more we average images, the better were the noise reduction effects and preservation of image details acquired.

## 6. Limitations

Although the proposed simple scheme is powerful, there are practical limitations that should be considered. First, noise extraction should be carefully and manually performed to include all noise patches so as to follow the characteristics of the source domain images, as they are critically related to the future performance of the visual inspection task. Second, although this scheme is powerful at high noise levels, restoring even more severely corrupted noisy images with indistinguishable noise and details was nearly impossible, as meaningful noise patches could not be extracted.

## 7. Conclusions

In this paper, we introduced a novel method to solve the unpaired heavy-noise SR task by targeting the images used in semiconductor visual inspection. A simple yet effective average denoising technique was applied to a heavy-noise dataset for noise reduction while preserving the structural information. In addition, to consider the noise component of the unpaired dataset, a paired dataset was constructed through noise patch extraction and injection with RealSR. Since there is no publicly available heavy-noise unpaired HR–LR dataset, we arbitrarily selected a ground-truth dataset and created a paired dataset; for a different ground-truth dataset, we can train it by applying the proposed scheme. Our empirical results were shown to be robust and to perform well against multiple noise levels, especially with extremely heavy-noise images; our method was also shown to be applicable to other SR networks.

## Figures and Tables

**Figure 1 sensors-22-08003-f001:**
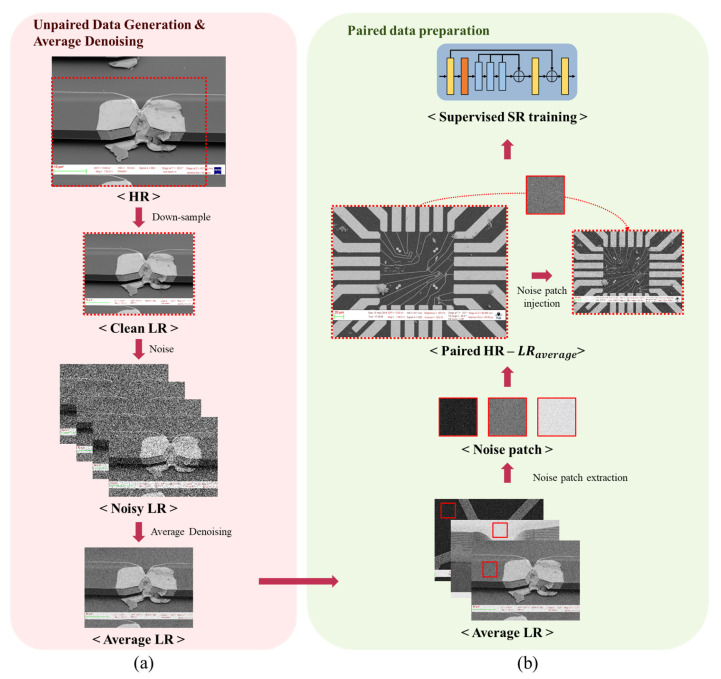
Overall architecture of the proposed method: (**a**) unpaired data generation and average denoising scheme, and (**b**) paired data generation.

**Figure 2 sensors-22-08003-f002:**
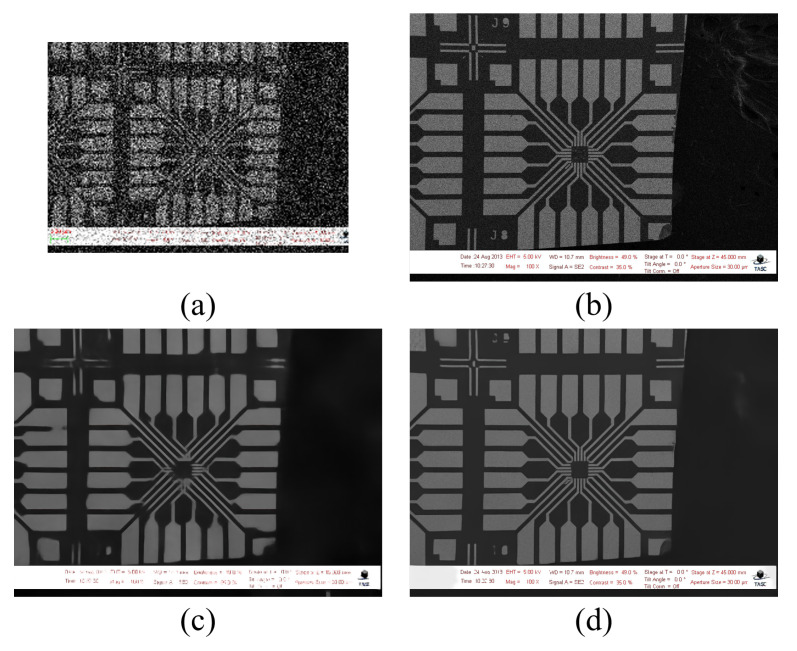
Limitations of applying naive SR methods: (**a**) input LR image, (**b**) target HR image, (**c**) output of naive inference of SwinIR method, and (**d**) output of proposed scheme on SwinIR.

**Figure 3 sensors-22-08003-f003:**
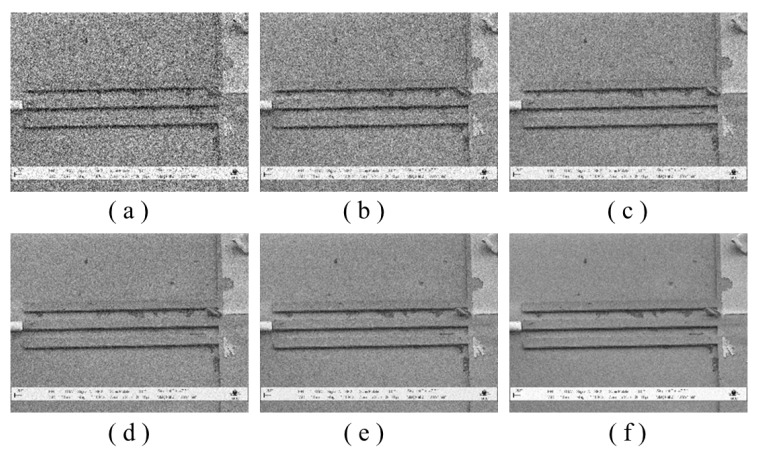
Effectiveness of noise averaging for heavy-noise images: (**a**–**e**) results of averaging 1, 2, 4, 8, and 16 LR images, and (**f**) clean LR image that is down-sampled from the HR image.

**Figure 4 sensors-22-08003-f004:**
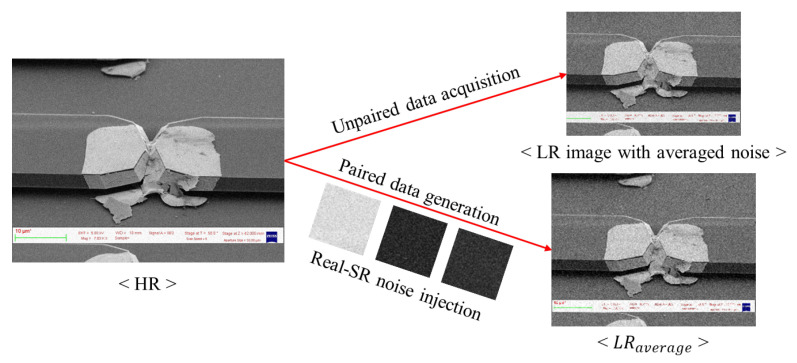
Data formulation: paired image generation from unpaired HR–LR dataset for supervised SR training.

**Figure 5 sensors-22-08003-f005:**
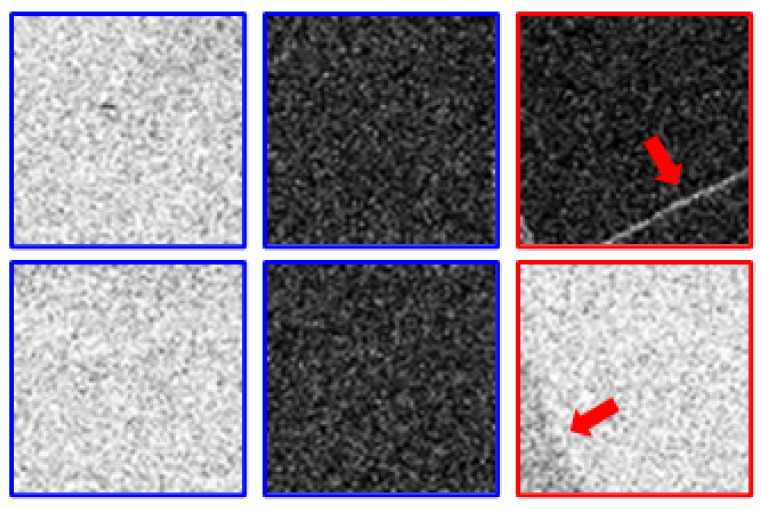
Noise patches extracted from the averaged output of four noisy LR images using RealSR [4]. The blue boxes indicate normal noise patches, and red boxes indicate falsely detected noise patches, with the arrows showing the falsely detected regions.

**Figure 6 sensors-22-08003-f006:**
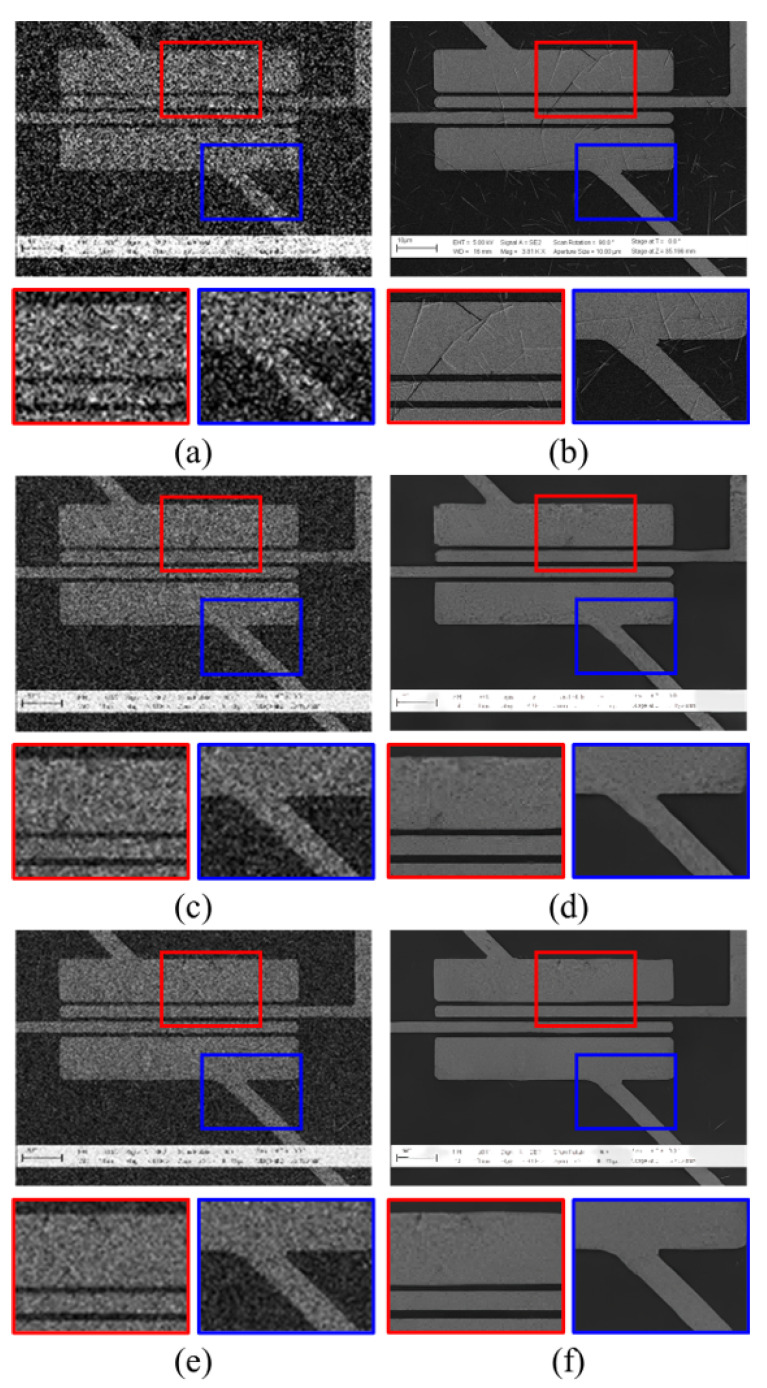
Qualitative comparison of the proposed scheme: (**a**) noisy LR image, (**b**) target HR image, (**c**,**e**,**g**) $LR_{average4}$, $LR_{average8}$, and $LR_{average16}$ input images, and (**d**,**f**,**h**) are SR results of (**c**,**e**,**g**) input $LR_{average}$ images, respectively. The baseline model SwinIR is used with a fixed noise level variance of 60 for fair comparison.

**Figure 7 sensors-22-08003-f007:**
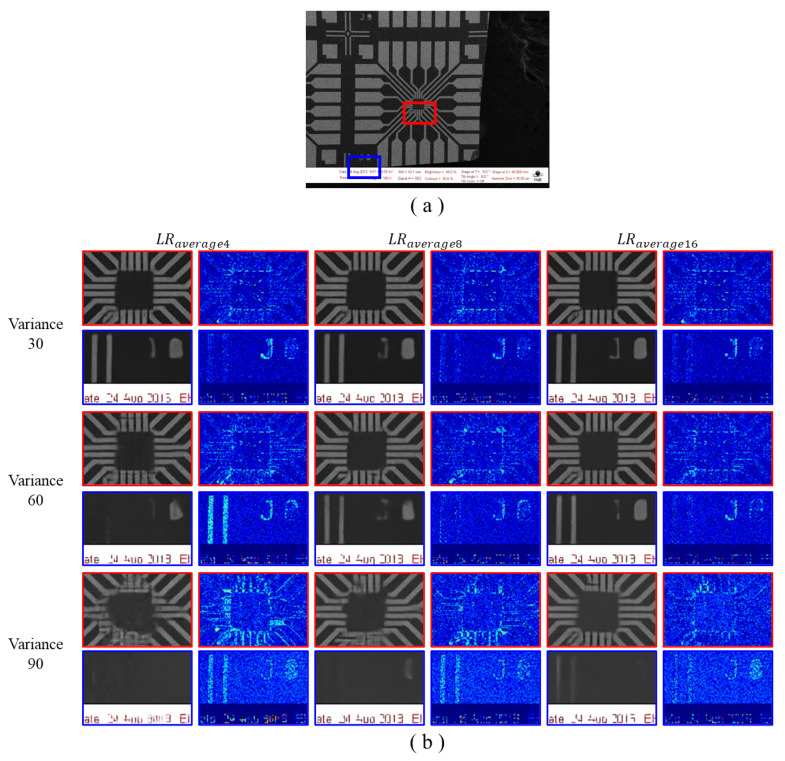
Qualitative comparison of our proposed scheme for different noise levels and different numbers of averaged images during denoising: (**a**) full-resolution image is the HR target image, which is given as a reference for the positions of the magnified boxes; (**b**) the magnified images and error map are illustrated in the heat map for reference (**a**). Images are evaluated for different noise levels (variances of 30, 60, and 90) and different numbers of averaged images for denoising (4, 8, and 16 images used). The red and blue boxes show the magnified SR results and the error map trained for each setting for $LR_{average}$ images at each noise level, respectively. The error map is illustrated as a heat map, with the error between the SR and target HR images amplified five times for visual comparison. The baseline network chosen is SwinIR.

**Figure 8 sensors-22-08003-f008:**
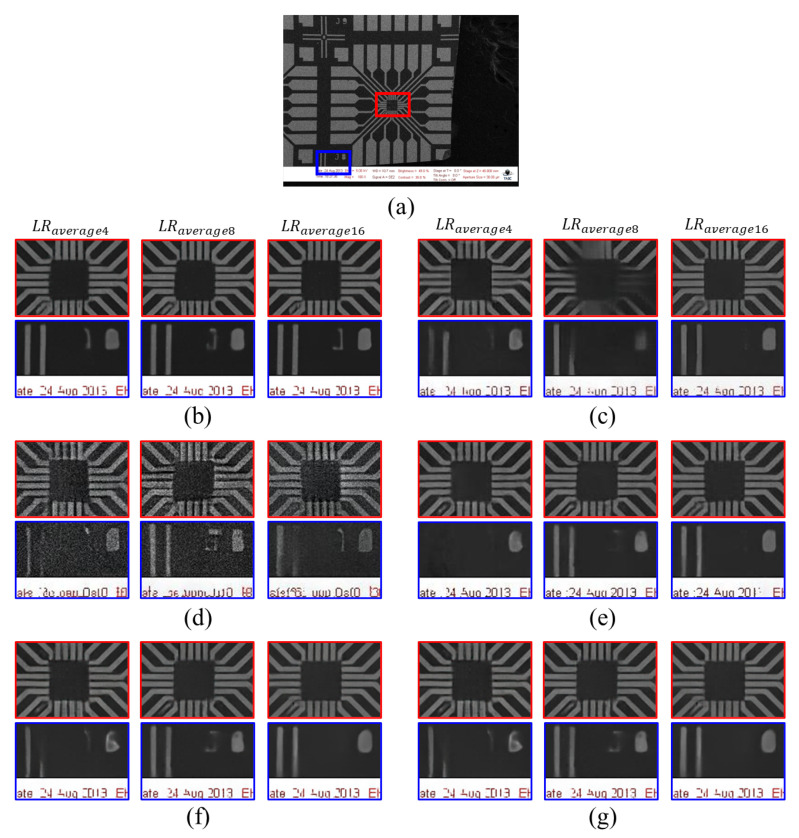
Qualitative comparison with conventional SR methods: (**a**) full-resolution image is the HR target image, which is given as a reference for the position of the magnified boxes; (**b**) SwinIR [26], (**c**) RCAN [9], (**d**) ESRGAN [12], (**e**) MSRResNet [11], (**f**) LTE [30], and (**g**) HAN [31]. For fair comparison, noise variance for $LR_{average}$ is set to 60.

**Figure 9 sensors-22-08003-f009:**
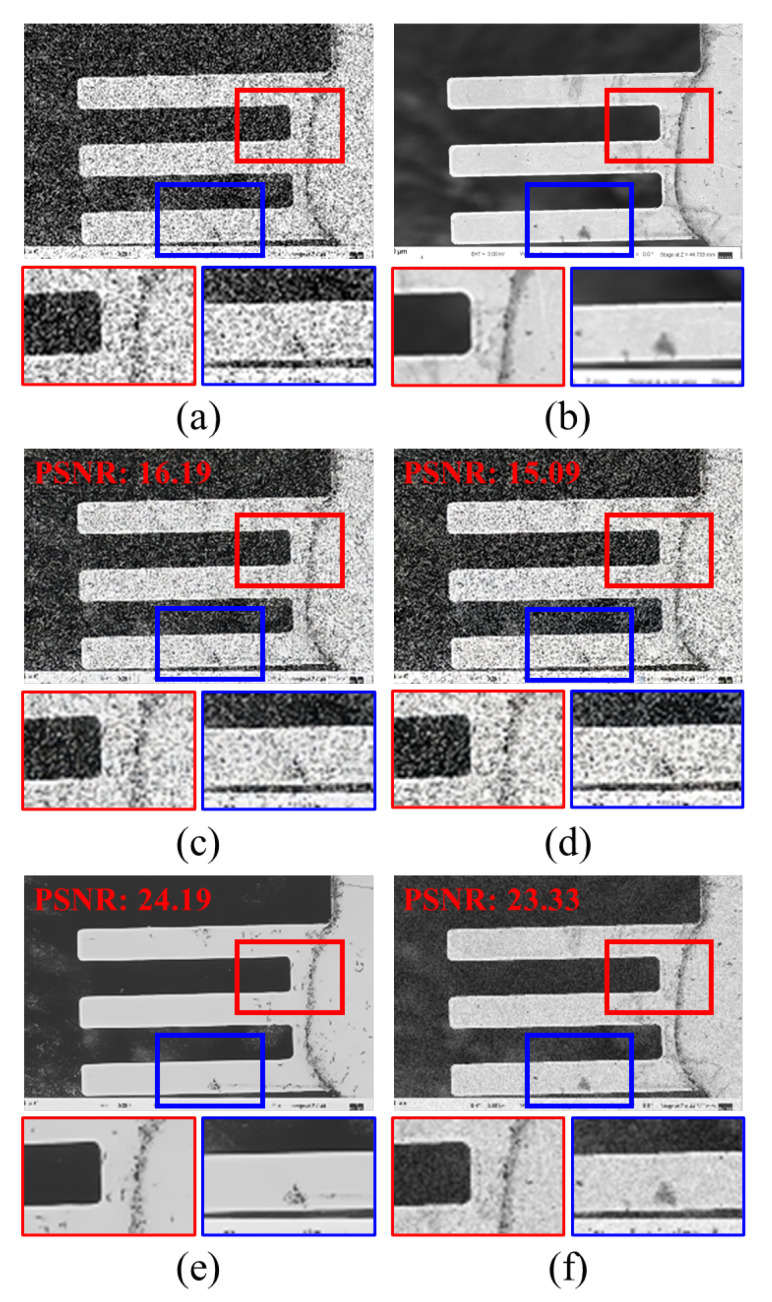
Visual comparison of other denoising algorithms: (**a**) input noisy LR image, (**b**) target clean LR image, and denoising by (**c**) FFDNet, (**d**) SwinIR, (**e**) Restormer non-blind model (trained with a fixed noise variance of 50), and (**f**) average denoising.

**Figure 10 sensors-22-08003-f010:**
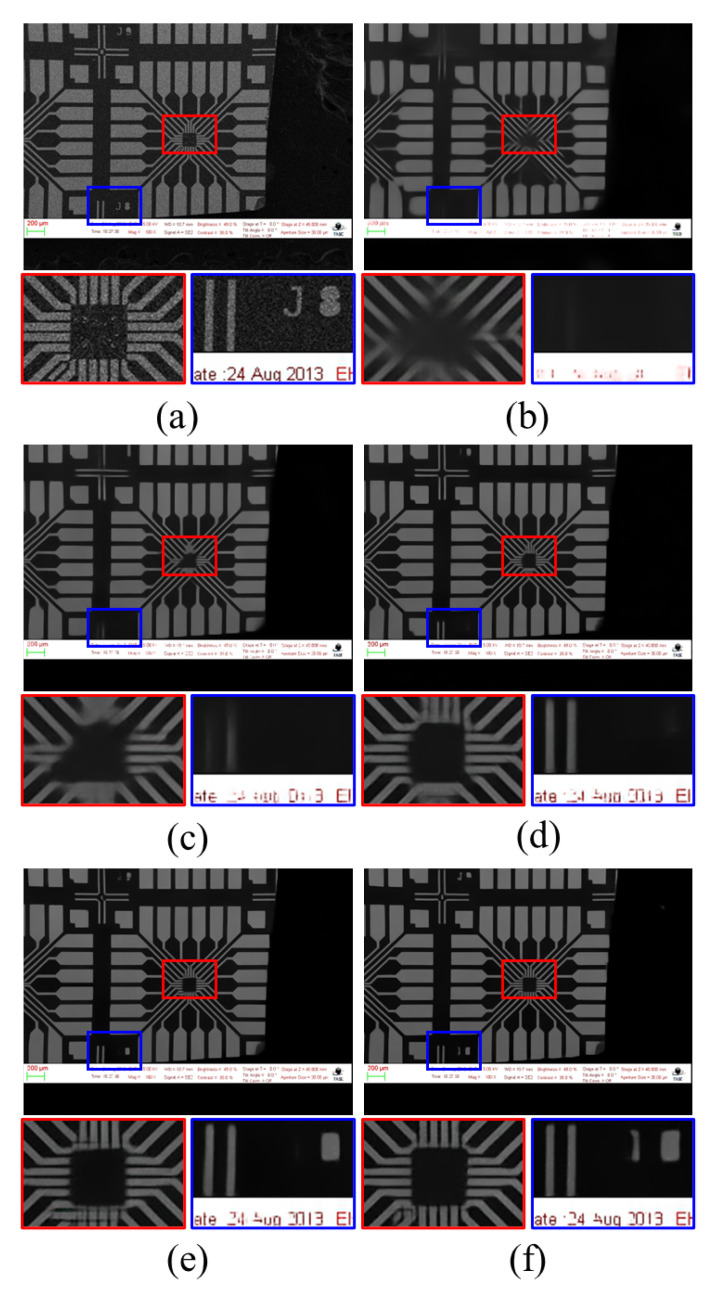
Visual comparison of average denoising by aligned dataset: (**a**) target HR image, (**b**–**f**) SR result by average denoised LR image using 1, 2, 4, 8, and 16 LR images averaged.

**Table 1 sensors-22-08003-t001:** Quantitative comparison of the proposed scheme on SwinIR [26] model based on PSNR, SSIM, LPIPS, and NIQE evaluations. PSNR and SSIM: higher is better; LPIPS and NIQE: lower is better.

		Full-Reference IQA									No-Reference IQA		
		PSNR (↑)			SSIM (↑)			LPIPS (↓)			NIQE (↓)		
	**Noise**	**30**	**60**	**90**	**30**	**60**	**90**	**30**	**60**	**90**	**30**	**60**	**90**
x4	$LR_{average4}$	24.9349	22.5349	20.6059	0.5188	0.4637	0.4083	0.5394	0.5466	0.5924	9.4209	10.5477	10.7917
	$LR_{average8}$	25.2433	22.9662	20.3939	0.5280	0.4847	0.4429	0.5385	0.5446	0.5897	9.0028	9.2508	9.9893
	$LR_{average16}$	25.3099	23.2101	20.6059	0.5340	0.4916	0.4555	0.5285	0.5395	0.5840	8.7917	8.9924	9.1307

**Table 2 sensors-22-08003-t002:** Quantitative comparison with conventional SR methods: SwinIR [26], RCAN [9], ESRGAN [12], MSRResNet [11], LTE [30], and HAN [31] on PSNR and SSIM. For fair comparison, noise variance for $LR_{average}$ is set to 60.

		SwinIR	RCAN	ESRGAN	MSRRESNET	LTE	HAN
x4	$LR_{average4}$	22.5349/0.4637	23.4808/0.5416	18.9535/0.3748	23.9163/0.5437	22.6516/0.4680	22.5060/0.4298
	$LR_{average8}$	22.9662/0.4847	24.0936/0.5569	20.1702/0.4062	24.1405/0.5615	22.8547/0.4793	23.3888/0.4872
	$LR_{average16}$	23.2101/0.4916	24.3732/0.5629	21.5123/0.5065	24.3646/0.5667	23.1209/0.4879	24.1599/0.5043

**Table 3 sensors-22-08003-t003:** Qualitative comparison of the proposed scheme on the aligned dataset depicted in Figure 10. Each evaluation was conducted using the SR result corresponding to the LR image with averaged noise. The top row depicts the number of images used for average denoising.

No. of LR Images	1	2	4	8	16
PSNR	21.4153	21.7353	22.0748	22.3320	22.6458
SSIM	0.4463	0.4417	0.4420	0.4483	0.4581
LPIPS	0.6347	0.6127	0.5988	0.5828	0.5678
NIQE	11.8199	11.2220	10.9082	10.4379	10.3903

## Data Availability

Publicly available datasets were analyzed in this study. Training and test datasets can be found here: https://b2share.eudat.eu/records/f1aa0f5ad38c456eaf7b04d47a65af53 (accessed on 3 September 2022) and https://b2share.eudat.eu/records/b9abc4a997f8452aa6de4f4b7335e582 (accessed on 3 September 2022).

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
