# Peer review of "Super-Resolving Methodology for Noisy Unpaired Datasets"

_sensors, 2022, doi:10.3390/s22208003_

Round 1

Reviewer 1 Report

In this paper, the authors introduced a novel method to solve the unpaired heavy-noise SR task by targeting the images used in the semiconductor process vision inspection. A simpleand   effective average denoising technique was applied to a heavy-noise dataset for noise reduction while preserving the structural information.

The paper is easy to follow. However, there are still some concerns on the current manuscript:

1.      The authors have to specify the new contributions in this work over the previous works.

2. Regarding face analysis, some important references are missing:

Face hallucination using multisource references and cross-scale dual residual fusion mechanism; International Journal of Intelligent Systems, 2022;Super-resolution: a comprehensive survey
Kamal Nasrollahi & Thomas B. Moeslund
Machine Vision and Applications volume 25, pages1423–1468 (2014)
Super-resolution optical imaging: A comparison,Micro and Nano Engineering  Volume 2, March 2019, Pages 7-28

3. The authors are suggested to specify their new contributions in this paper, as well as mention the disadvantage and drawbacks of the existing methods and clarify how their proposed model can overcome them.

4. Currently, comparative evaluation is weak.

As a journal paper, the proposed method also lacks of comparisons to other state-of-the-art algorithms (e.g. published after 2020).

Author Response

We appreciate your constructive and valuable comments. We revised the manuscript very carefully to accommodate all of the concerns raised by the reviewers. We have made our best efforts to follow the reviewers’ comments while keeping the key contents of the original version. The major improvements are as follows:

  • We performed additional experiments and added corresponding results.
  • We modified figures, tables, and some sentences to clarify the contribution for this manuscript.
  • To improve the readability, we added some sentences and references.

We enclose the reply letters to reviewers’ comments.

Reviewer 2 Report

This paper is well written. The following are the suggestions to be incorporated in the final version:

1. Test images are not enough, add more test sample images and retry the experiment.

2. There are some miss-types problems.

3. Figure caption should be more descriptive, for example figure 5 is not clear to understand it's description.

4. The text in the figures should be enlarged and clear.

Author Response

(The authors gave the same response as above.)

Round 2

Reviewer 1 Report

Accepted